# Gomisin M2 Ameliorates Atopic Dermatitis-like Skin Lesions via Inhibition of STAT1 and NF-κB Activation in 2,4-Dinitrochlorobenzene/*Dermatophagoides farinae* Extract-Induced BALB/c Mice

**DOI:** 10.3390/molecules26154409

**Published:** 2021-07-21

**Authors:** Jinjoo Kang, Soyoung Lee, Namkyung Kim, Hima Dhakal, Taeg-Kyu Kwon, Eun-Nam Kim, Gil-Saeng Jeong, Sang-Hyun Kim

**Affiliations:** 1Department of Pharmacology, School of Medicine, Kyungpook National University, Daegu 41566, Korea; jinjoo1kang@gmail.com (J.K.); nortonnklab@gmail.com (N.K.); dhakalhima@gmail.com (H.D.); 2Immunoregulatory Materials Research Center, Korea Research Institute of Bioscience and Biotechnology, Jeongeup 28116, Korea; sylee@kribb.re.kr; 3Department of Immunology, School of Medicine, Keimyung University, Daegu 42601, Korea; kwonth@dsmc.or.kr; 4College of Pharmacy, Keimyung University, Daegu 42601, Korea; enkimpharm@gmail.com

**Keywords:** atopic dermatitis, gomisin M2, *Dermatophagoides farinae* extract, keratinocytes, 2,4-dinitrochlorobenzene

## Abstract

The extracts of *Schisandra chinensis* (Turcz.) Baill. (Schisandraceae) have various therapeutic effects, including inflammation and allergy. In this study, gomisin M2 (GM2) was isolated from *S. chinensis* and its beneficial effects were assessed against atopic dermatitis (AD). We evaluated the therapeutic effects of GM2 on 2,4-dinitrochlorobenzene (DNCB) and *Dermatophagoides farinae* extract (DFE)-induced AD-like skin lesions with BALB/c mice ears and within the tumor necrosis factor (TNF)-α and interferon (IFN)-γ-stimulated keratinocytes. The oral administration of GM2 resulted in reduced epidermal and dermal thickness, infiltration of tissue eosinophils, mast cells, and helper T cells in AD-like lesions. GM2 suppressed the expression of IL-1β, IL-4, IL-5, IL-6, IL-12a, and TSLP in ear tissue and the expression of IFN-γ, IL-4, and IL-17A in auricular lymph nodes. GM2 also inhibited STAT1 and NF-κB phosphorylation in DNCB/DFE-induced AD-like lesions. The oral administration of GM2 reduced levels of IgE (DFE-specific and total) and IgG2a in the mice sera, as well as protein levels of IL-4, IL-6, and TSLP in ear tissues. In TNF-α/IFN-γ-stimulated keratinocytes, GM2 significantly inhibited IL-1β, IL-6, CXCL8, and CCL22 through the suppression of STAT1 phosphorylation and the nuclear translocation of NF-κB. Taken together, these results indicate that GM2 is a biologically active compound that exhibits inhibitory effects on skin inflammation and suggests that GM2 might serve as a remedy in inflammatory skin diseases, specifically on AD.

## 1. Introduction

Atopic dermatitis (AD) is a chronic inflammatory skin disease related to an immune-mediated response and is characterized by several factors including skin redness, dryness, pruritus, hyperplasia of keratinocytes, infiltration of immune cells, and T-cells expansion of the lymph node [1,2]. With respect to the changes in histopathology in AD lesions, the changes include the thickening and swelling of the lesion area and an increased number of fully granulated mast cells, eosinophils, and infiltrated lymphocytes [3]. Immunologically, high titers of antibody production are induced with an abnormal Th2-cell immune response, which acts to activate B cell proliferation and class-switching [4,5]. One of the main responses to type 2-mediated inflammation is the infiltration of eosinophilic and basophilic cells to the skin and an extensive mast cell degranulation occurs in a process that is dependent on cross-linking of surface-bound IgE [6].

The AD therapeutic agents that are most widely used include corticosteroids, antihistamines, and immunosuppressive drugs [7]. However, their continued use can cause a variety of adverse effects [7,8]. Therefore, the interest in identifying natural compounds such as alternative medicines to existing treatments is increasing due to their safety and activity on the immune system [9].

Extensive use of *Schisandra chinensis* (Turcz.) Baill. (Schisandraceae) has been adopted in traditional Chinese medicine and Western herbal medicines and is widely used for the treatment of asthma, rheumatism, and arthritis [10,11]. In particular, the fruits of *S. chinensis* have exhibited a diverse range of pharmacological effects, such as having antiallergic, anti-inflammatory, and antiviral properties [12,13]. There is a multitude of methods that *Schisandra* can be used as a supplement, which includes the forms of dried powder, pills, extracts, and elixirs due to its association with health and immunity. The extracts of *S. chinensis* have active lignans, such as deoxyschisandrin, γ-schisandrin, and gomisin [14,15]. Among the lignans, gomisin M2 (GM2) has slightly greater antiallergic properties through the suppression of mast cell activation compared to the other lignans [16]. Although whole *S. chinesis* extracts have been reported to suppress AD [17,18], the biological and pharmacological activities of GM2 within *S. chinesis* extracts are not well characterized as chronic relapsing inflammatory skin disease, particularly AD. Therefore, the alleviation of atopic inflammatory response to AD is expected with GM2 due to its the anti-inflammatory effects. Herein, this study aimed to elucidate the effects of GM2 on the AD-like skin inflammation and to define the molecular mechanisms of these effects.

## 2. Results

### 2.1. The Effects of GM2 on the Phenotypic Characteristics of DNCB/DFE-Induced AD Lesions

GM2 was isolated as detailed in the flow diagram of Figure 1A and the chemical structure is shown in Figure 1B. Next, in order to investigate the effect GM2 has on AD, 2,4-dinitrochlorobenzene (DNCB)/*Dermatophagoides farinae* extract (DFE)-induced AD mice model was utilized as detailed in the experimental protocol (Figure 1C). During the induction period of AD, the body weights of GM2 mice group were not altered, which indicates that no toxicity was present (Appendix A). In order to determine whether GM2 suppressed the inflammatory phenotype, we histologically examined the ear images and ear sections. In the ears of DNCB/DFE-induced AD mice group, inflammatory skin damage such as redness, keratinization, and increased thickness were present compared to the control group. However, oral administration of GM2 decreased in the pathological progressions (Figure 2A) and ear thickness in a dose-dependent manner (Figure 2B). Upon the histologic examination of the ear tissues, a significant decrease in epidermal and dermal thickness of the ears by GM2 was observed compared to the AD group (Figure 2C).

### 2.2. The Effects of GM2 on the Infiltration of Inflammatory Cells in DNCB/DFE-Induced AD Lesions

Swelling is caused by the infiltration of inflammatory cells that are involved in a hyperkeratosis, parakeratosis, and acanthosis [3]. Therefore, in order to analyze the effects of GM2 on the inflammatory cell infiltration in the ear tissue, they were stained with hematoxylin and eosin (H&E) and toluidine blue (TB). The infiltration of the tissue eosinophils was assessed by H&E-staining and mast cells were observed with TB-staining in AD lesions. In the DNCB/DFE-induced AD mice, the infiltration of the tissue eosinophils and mast cells in the AD lesions was observed; however, oral administration of GM2 resulted in a reduction in those infiltrations (Figure 3A,B). During the pathologic process of skin, CD44 is expressed on keratinocytes membranes and the infiltrating lymphocytes near the inflammation localization or tumor localization [19,20]. Therefore, in order to confirm the role of GM2 on cell infiltration, we evaluated the expression of CD4 and CD44 in the ear tissues of DNCB/DFE-induced AD. Oral administration of GM2 (10 mg/kg) suppressed the expressions of CD4 and CD44 in the AD-like skin (Figure 3C).

### 2.3. The Effects of GM2 on the Inflammatory Mediators in DNCB/DFE-Induced AD Lesions

In order to investigate the actions of GM2 on the gene expression of cytokines within the DNCB/DFE-induced AD-like skin inflammation, we examined the expression of the inflammatory cytokines IL-1β, -4, -5, -6, -12a, and TSLP. In comparison to the AD mice, the expression of these gene decreased after oral administration of GM2 (Figure 4A).

Next, we evaluated whether GM2 suppresses Th-related cytokines in the lymph nodes of the DNCB/DFE-induced AD mice by measuring interferon (IFN)-γ, IL-4, and IL-17A, which are representative of Th1, Th2, and Th17 cells. The expression of IFN-γ, IL-4, and IL-17A increased in draining lymph nodes (dLNs) of AD mice, however, these cytokines were reduced by the oral administration of GM2 (Figure 4B).

For the responsible mechanisms of GM2, inhibitory effects were evaluated through the analysis of activated signal transducer and activator of transcription 1 (STAT1) and nuclear factor (NF)-κB in the ear tissues. It was observed that the oral administration of GM2 inhibited the phosphorylation of STAT1 and NF-κB p65 in the ear tissues of AD mice (Figure 4C).

### 2.4. The Effects of GM2 on AD-Related Immunoglobulin and Protein Levels in DNCB/DFE-Induced AD Mice

The role of GM2 in Th1-associated and Th2-associated immune responses in AD mice was estimated using the serum levels of DFE-specific IgE, total IgE, and IgG2a. In the DNCB/DFE-induced AD mice, increased levels of DFE-specific IgE, total IgE, and IgG2a were observed. However, the oral administration GM2 group significantly reduced the serum levels of IgE (DFE-specific and total) and IgG2a (Figure 5A). In addition, the representative cytokines IL-4, IL-6, and TSLP, which are important to the progression of AD, were reduced by GM2 in the ear tissue compared to the AD group (Figure 5B).

### 2.5. The Effects of GM2 on the Activated Keratinocytes

Keratinocytes (HaCaT cells) were utilized to confirm the role of GM2 in vitro. First, the cytotoxicity of GM2 was evaluated using an MTT assay. GM2 did not show cytotoxicity up to 10 μM (Appendix A). Therefore, subsequent in vitro assays utilized GM2 up to a concentration of 10 μM. GM2 (0.1, 1, or 10 μM) decreased the expression of the inflammatory cytokines (IL-1β and IL-6) and the chemokines (CXCL8 and CCL22) in comparison to the tumor necrosis factor (TNF)-α/IFN-γ-stimulated HaCaT cells (Figure 6A). STAT1 and NF-κB are important transcription factors that regulate the expression of those inflammatory mediators [21,22]. In order to evaluate the responsible mechanisms in the reduction in cytokine/chemokine, we assessed the effects of GM2 on the activation of STAT1 and NF-κB p65. GM2 suppressed the phosphorylation of STAT1, degradation of IκBα, and nuclear translocation of NF-κB p65 in the TNF-α/IFN-γ-stimulated HaCaT cells (Figure 6B). 

## 3. Discussion

AD patients are widely treated using broad T-cell targeting medicines for the rebalancing between Th1-cells and Th2-cells [23]. Commonly used therapeutic agents for AD include corticosteroids, but these have various adverse effects that include skin thinning, atrophy, and kidney failure [7,8]. Therefore, we aimed to identify a suitable candidate derived from natural compounds that have proven efficacy in the treatment of skin diseases for this study.

The European Food Safety Authority has accepted that fruit from *S. chinensis* is a safe nutrition ingredients with applications in the treatment of mental well-being since 2010 [24]. *S. chinensis* inhibits immunoglobulin, immune cell infiltration, and air hyper-responsiveness with potent anti-asthmatic activity in an asthma mouse model [10,25]. GM2 can be found in the fruit extracts of both *S. rubriflora* [26] and *S.*
*chinensis* (Turcz.) [13]. To date, it has been reported that GM2 was isolated from the fruit of *S. rubriflora* had an anti-HIV replication effect on H9 lymphocytes in vitro [26]. In addition, we have previously reported that GM2 isolated from *S. chinensis* fruit inhibited acute inflammation caused by mast cell-mediated allergic inflammation [16]. In this study, we have defined the role of GM2 in the chronic inflammatory skin disease, in particular, AD.

The causes of chronic AD are exacerbated by several external factors, such as antigen, irradiation, microorganisms, and allergens [27,28]. A number of characteristics of chronic AD lesions include dryness, thickened skin, and lichenification as a result of the immune response. The histopathological features of AD eventually expand the epidermal and dermal thickness through the infiltration of inflammatory cells including mast cells, tissue eosinophils, and T-cells [3]. Additionally, the skin symptoms of AD are induced by the release of inflammatory cytokine and IgE-mediated cell activation [29,30]. AD-like mice produced a variety of cytokines, including the pro-inflammatory cytokine IL-1β; Th2-associated cytokines IL-4, IL-5, IL-6, and TSLP; and the Th1-associated IL-12a. All of which are important players in the pathogenesis of allergic reactions [31]. Therefore, we demonstrate an inhibitory effect of GM2 on these AD-related symptoms and we postulate that GM2 suppressed the inflammatory response by decreasing the activities of the immune cells involved in the development of AD.

The activated antigen-presenting cells in inflammatory AD skin migrate to the lymph nodes as well as prime naïve T-cells into Th2 cells. Then, these specific T-cells are activated and respond to the production of cytokines and the activation of inflammatory cells [32,33]. Moreover, it has been shown in previous studies that IL-4 induces IgE production and IFN-γ induces the production of IgG2a [34]. Therefore, we assessed the expression of Th cell signature cytokines in dLNs of mice to determine whether GM2 regulates T-cell differentiation. Subsequently, we evaluated whether GM2 regulates the levels of sera immunoglobulin in the DNCB/DFE-induced AD-like mouse model. The result demonstrated that GM2 suppressed the Th-cell-mediated immune response in dLNs and sera of AD mice. In addition, we observed that the oral administration of GM2 was able to suppress the release protein levels of IL-4, IL-6, and TSLP of AD-like lesions. Therefore, we suggest that GM2 possesses a suppressive effect on the immune response, which may be utilized in the treatment of AD.

A variety of phenotypes and immune responses are observed differently for lesions and non-lesions in AD due to keratinocytes and T cells [35]. In AD pathogenesis, keratinocytes have a similar effect on the immune response of AD through the stimulation of TNF-α and IFN-γ [3,36]. Therefore, in this study, the effects of GM2 were investigated using keratinocytes (HaCaT cells) in vitro. Among the cytokines and chemokines, the transcription of CXCL8, CCL22, and IL-6 is modulated through an NF-κB element and IL-6 activates the tyrosine phosphorylation of STATs [37,38,39]. We observed that treatment with GM2 suppressed the gene expression of IL-1β, IL-6, CXCL8, and CCL22 in TNF-α/IFN-γ-stimulated HaCaT cells through the inhibition of the STAT1 phosphorylation and the nuclear translocation of NF-κB. Therefore, we suggest that GM2 may have an inhibitory effect on the expression of inflammatory cytokines and chemokines through the signal inhibition of the STAT1 and NF-κB. Nevertheless, further experiments are required to explore the long-term effects of GM2 in AD-like skin inflammation.

## 4. Materials and Methods

### 4.1. Extraction, Isolation, and Identification of GM2

The fruits of *S. chinensis* (Turcz.) Baill. was purchased from the Yangnyeong herbal medicine market (Daegu, Korea) and were identified by Prof. Jeong of the College of Pharmacy, Keimyung University, Korea, and a voucher specimen (No. KPP2018-1022) was deposited. A total of 20 kg of *Schisandra chinensis* (Turcz.) Baill. fruits underwent extraction with 95% EtOH (10 L) at room temperature for 5 days. The EtOH extract evaporated in a vacuum that yielded 5.7 kg of residue, which was subsequently suspended in H_2_O and successively partitioned with CH_2_Cl_2_, EtOAc, and *n*-BuOH. The CH_2_Cl_2_ extract (525 g) was applied to a silica gel column chromatography with a gradient of ether/acetone (20:1 to 1:2) to the isolation of five major fractions (Fr. 1–Fr. 5). Fr. 1 was subjected to Sephadex LH-20 elution with MeOH/H_2_O (1:1), which yielded two fractions (Fr.1-1 and Fr. 1-2). Fr. 1-2 underwent medium-pressure liquid chromatography and eluted with MeOH/H_2_O (10:1 to 1:1), which yielded compound **2** (64 mg, yield 0.012%). The isolated Fr. 1-2 was identified as GM2 utilizing ^1^H and ^13^C-NMR and their spectral data were compared with previously published data [40,41].

GM2: HRESIMS *m*/*z*: 387 [M+H]^+^; ^1^H NMR (CDCl3, 500 MHz): δH 6.45 (H-11), 5.93 (1-H, d, OCH2O), 3.80 (3-H, s, OMe-12), 3.57 (3-H, s, OMe-1), 3.49 (3-H, s, OMe-13), 2.42 (1-H, dd, J = 13.4, 7.7, H-9), 2.21 (1-H, dd, J = 13.4, 1.9, H-9), 1.98 (1-H, dd, J = 13.1, 9.3, H-6), 0.93 (3-H, d, J = 7.3, H-17), and 0.70 (3-H, d, J = 7.0, H-18); ^13^C NMR(CDCl3, 500MHz): δC 149.6 (C-12), 147.9 (C-3), 147.5 (C-14), 139.2 (C-1), 136.9 (C-5), 134.1 (C-2), 133.6 (C-13), 133.0 (C-10), 121.0 (C-16), 117.0 (C-15), 106.1 (C-11), 103.2 (C-4), 100.7 (OCH2O), 59.7 (OMe-13), 58.2 (OMe-1), 55.1 (OMe-12), 40.7 (C-7), 38.4 (C-9), 35.3 (C-6), 33.2 (C-8), 21.8 (C-17), and 12.8 (C-18).

### 4.2. Reagents

All reagents were purchased from Sigma-Aldrich (St. Louis, MO, USA) unless stated otherwise. Both recombinant human TNF-α and IFN-γ were purchased from R&D Systems (Minneapolis, MN, USA). The DFE powder was purchased from Prolagen (Seoul, Korea), and 0.5% Tween 20 in phosphate-buffered saline (PBS) was added to this extract. In order to dissolve the DNCB, a mixture of acetone and olive oil (3:1, *v*/*v*) was added. Primers used for qPCR and the antibodies used for Western blot are described in Appendix A.

### 4.3. Ethics Statement and Cell Maintenance

All procedures used in the animal experiments were run in accordance with the guidelines established by the Public Health Service Policy on the Humane Care and Use of Laboratory Animals and all experiments were approved by the Institutional Animal Care and Use Committee of Kyungpook National University (IRB# 2021-0073). Five-week-old female BALB/c mice (*n* = 35) were purchased from Dae Han Experimental Animal Center (Daejeon, Korea). The animals (*n* = 5/cage) were provided with food and water ad libitum in a laminar airflow room maintained at 22 ± 2 °C, with a relative humidity of 55 ± 5% and a 12 h light/dark cycle throughout the study.

The keratinocytes (HaCaT cell line) were maintained in Dulbecco’s modified Eagle’s medium (Gibco, Grand Island, NY, USA) that was supplemented with 10% fetal bovine serum (Gibco) and antibiotics (100 U/mL penicillin G, 100 μg/mL streptomycin, Gibco). Cells were cultured at 37 °C in 5% CO_2_.

### 4.4. Induction of AD-like Skin Inflammation in the Mouse Ears and Administration of GM2

A total of 35 mice were divided into seven groups (*n* = 5): vehicle (PBS), GM2 10 mg/kg, DNCB/DFE and vehicle (PBS), DNCB/DFE plus GM2 (0.1, 1, or 10 mg/kg), or DNCB/DFE plus Dexa 1 mg/kg. During the first week of induction, DNCB (2%, 20 μL/ear) was applied onto each ear twice for sensitization. Then, DNCB (1%, 20 μL/ear, once) and DFE (1 mg/mL, 20 μL/ear, twice) were applied to both BALB/c mouse ears for three weeks. After two weeks, GM2 (0.1, 1, or 10 mg/kg) or Dexa (1 mg/kg) were orally administrated by gavage for five consecutive days per week for two weeks.

### 4.5. Preparation of Animal Experimental Samples

During the experiment period, the ear thicknesses were measured using a 7301-dial thickness gauge (Mitutoyo, Co., Tokyo, Japan) at 24 h after DNCB or DFE application.

Mice were euthanized by carbon dioxide on day 28 and whole blood was collected through the abdominal vena cava. The blood samples were centrifuged (GYROZEN Co., Ltd., Seoul, Korea) at 400× *g* for 15 min and stored at −70 °C for use when required for serum enzyme-linked immunosorbent assay (ELISA). The auricular lymph nodes (draining lymph nodes; dLNs) were removed and processed for qPCR analysis. Ear tissues were removed and used for histological observation, ELISA, and Western blot analysis.

### 4.6. Histological Analysis

Mice ears used in this experiment were fixed in 10% formaldehyde and embedded. Each tissue was sectioned with 6 µm thickness and subsequently stained H&E and TB. Epidermal and dermal thickness, tissue eosinophils, and mast cells were all observed using a Carl Zeiss microscope (Jena, Germany). The epidermal and dermal thickening was analyzed on the stage micrometer 10:100 (Zeiss) on the H&E-stained tissue slides at 100× magnification. Five randomly selected sites were measured for thickness from each slide. The infiltration of tissue eosinophils and mast cells in the H&E-stained and TB-stained tissues were counted from five randomly-selected sites from each tissue slide at a high-power field (HPF) and images were acquired using a Carl Zeiss microscope at 400× magnification.

### 4.7. Quantitative Real-Time Polymerase Chain Reaction (qPCR)

The HaCaT cells (2 × 10^5^ cells/24-well plates) were pretreated with either GM2 (0.1, 1, or 10 μM) or Dexa (10 μM) for 1 h and then stimulated with TNF-α/IFN-γ (10 ng/mL) for 6 h for RNA extraction. On the other hand, the ears and dLNs of DNBC/DFE-induced AD-like mice were prepared at the end of the experimental period before tissue samples were homogenized by a TissueLyser II (Qiagen, Hilden, Germany). Total RNA was isolated using an RNAiso Plus kit (Takara Bio Inc., Shiga, Japan) and the quantification of the RNA concentrations was performed using a NanoDrop 2000 spectrophotometer (ThermoFisher scientific). Complementary DNA (cDNA) was prepared using a RevertAid RT kit (ThermoFisher Scientific, Wilmington, DE, USA). The qPCR assays were performed using Applied Biosystems StepOnePlus™ Real-Time PCR systems (Life Technologies Corporation, Kallang Avenue, Singapore) according to the manufacturer’s instructions. The cycle number was optimized to ensure that the product accumulation was in the exponential range using a 20 μL reaction mixture. The master mix consisted of 2 μL of cDNA (200 ng, keratinocytes; 20 ng, ear tissues; 2 ng, dLNs), 1 μL of forward and reverse primers (0.4 μM), 10 μL of QGreenBlue Master Mix High ROX (QBHR-05 2X, ThermoFisher Scientific), and 6 μL of dH_2_O. The qPCR assays were normalized to GAPDH for HaCaT cells and β-actin was used for the tissue samples. Quantification analysis was performed using the StepOnePlus™ Real-Time PCR systems software version 2.3 as supplied by the manufacturer.

### 4.8. Western Blot

The HaCaT cells (1 × 10^6^ cells/6-well plate) were pretreated with either GM2 (0.1, 1, or 10 μM) or Dexa (10 μM) for 1 h, stimulated with TNF-α/IFN-γ (10 ng/mL) and then subsequently incubated for 15 min. The cells were washed with 1 mL of ice-cold PBS containing Na_3_VO_4_ and subsequently scraped with a lysis buffer (0.5 M Tris (pH 7.5), 5 M NaCl, 0.5 M EDTA, 10% glycerol, 1% Triton X-100, 0.1 M DTT, 1 mM Na_3_VO_4_). From the collected cells, the total cell lysates were obtained using sonication for 30 s and then centrifugation was performed at 2500× *g* at 4 °C for 20 min. The cytosolic proteins were obtained by centrifuged cells at 2000× *g* at 4 °C for 5 min and then the supernatant was used for the cytosolic protein extraction. The pellets were washed with 1 mL of PBS containing Na_3_VO_4_ and resuspended in 30 μL of ice-cold RIPA buffer (Biosesang, Gyonggi-do, Korea). Samples were left on ice for 20 min, vortexed, and then centrifuged at 16,000× *g* at 4 °C for 20 min, the resultant supernatant was used for the nuclear protein extraction. On the other hand, the ear tissue in lysis buffer (50 mM Tris-Cl (pH 7.4), 1 mM sodium ethylene diamine tetra-acetate, 1% Triton X-100, 0.2% sodium deoxycholate, 0.2% sodium dodecyl sulfate (SDS)) was homogenized by a TissueLyser II (Qiagen). All the lysis buffers contained protease/phosphatase inhibitor cocktail (Roche, Mannheim, Germany). Protein concentrations were measured using the Bradford method using a colorimetric assay dye (Bio-Rad Protein assay dye). The samples were electrophoresed in a 10% SDS-polyacrylamide gel and then transferred onto a BioTrace™ NT membranes (0.2 μm, Pall Corporation, Ann Arbor, MI). Membranes were blocked with 3% bovine serum albumin in Tris-buffered saline containing Tween 20 and the membranes were developed using a G:box Chemi XRQ (SYNGENE, Cambridge, UK) through a SuperSignal West Pico chemiluminescent substrate (ThermoFisher Scientific).

### 4.9. ELISA

According to the manufacturer’s instructions, the levels of serum IgE (DFE-specific and total), serum IgG2a, tissue extracted IL-4, and IL-6 were measured using BD Biosciences (Oxford, UK) and tissue extract TSLP was measured using the R&D System (Minneapolis, MN). For the detection of DFE-specific IgE, each well was coated with 25 μg/mL of DFE in PBS. In order to measure the protein levels of IL-4, IL-6, and TSLP, mice ear tissue was homogenized using a TissueLyser II (Qiagen) in an extraction buffer (100 mM Tris (pH 7.4), 150 mM NaCl, 1 mM EGTA, 1 mM EDTA, 1% Triton X-100, 0.5% sodium deoxycholate) that contained a phosphatase/protease inhibitor cocktail (Roche) and phenyl methyl sulfonyl fluoride. Debris was removed by centrifugation at 20,000× *g* for 15 min and then supernatants were collected. The determination of protein concentration was measured by the Bradford assays (colorimetric assay dye, Bio-Rad, Hercules, CA, USA). The absorbance was measured at 450 nm using a spectrophotometer (VersaMax™ Microplate Reader, Biocompare, Billerica, MA, USA). The data were analyzed and calculated using the SoftMax Pro software version 6 (Biocompare, Billerica, MA, USA).

### 4.10. Statistical Analysis

All data were statistically analyzed using Prism statistical software 6 (GraphPad Software, San Diego, CA, USA). The results are expressed as the mean ± SEM. The effect of the treatment was analyzed using a one-way analysis of variance followed by Dunnett’s multiple comparisons test. Differences were considered statistically significant where the *p*-value was *<* 0.05.

## 5. Conclusions

In this study, the oral administration of GM2 improved the symptoms of AD and inhibited the activation of the inflammatory mediators in the DNCB/DFE-induced AD-like mouse model. In addition, this study demonstrated that GM2 had anti-inflammatory activities through the inhibition of STAT1 and NF-κB signaling pathway in TNF-α/IFN-γ-stimulated keratinocytes. Therefore, we propose that GM2 extracted from *S. chinensis* is a potential candidate compound for the treatment of chronic inflammatory skin diseases, such as AD and psoriasis. Moreover, it is possible that eating the fruit of *S. rubriflora* or *S. chinensis* containing GM2 as a food supplement may be utilized to prevent chronic inflammation.

## Figures and Tables

**Figure 1 molecules-26-04409-f001:**
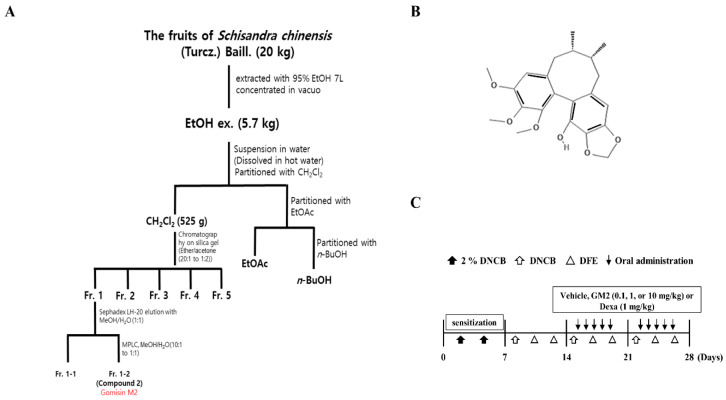
Isolation scheme, structure of GM2, and experimental scheme of AD induction. (**A**) Schematic diagram of GM2 isolation of the bioactive compounds from *Schisandra chinensis* (Turcz.) Baill. (*Schisandraceae*). (**B**) Chemical structure of GM2. (**C**) Experimental scheme for AD induction to application 2,4-dinitrochlorobenzene (DNCB) and *Dermatophagoides farina* (DNCB/DFE) and oral administration of gomisin M2 (GM2) or dexamethasone (Dexa).

**Figure 2 molecules-26-04409-f002:**
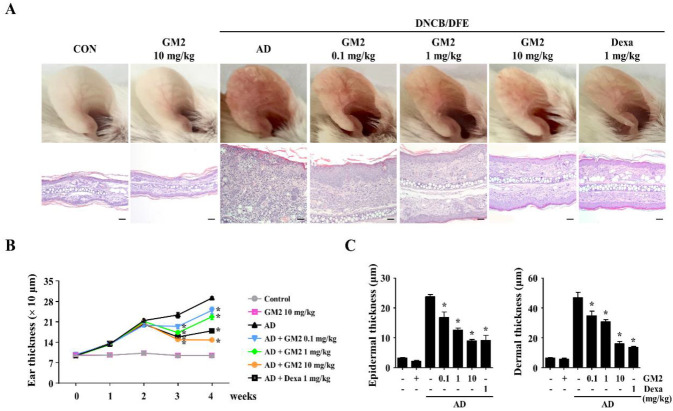
Representative atopic dermatitis (AD) lesion images and effects of GM2 on the ear thickness of AD mice. (**A**) Representative images demonstrated AD-like skin lesions (upper panel) in the different experimental mice groups and representative photomicrographs of the ear sections were stained with H&E (down panel, scale bar = 20 μm, magnification 200×) from the DNCB/DFE-induced mouse model. (**B**) Ear thickness was measured using a 7301-dial thickness gauge. (**C**) The epidermal and dermal thickness was measured using the stage micrometer 10:100 in the H&E-stained tissue slides. Data are presented in the graph and are the means ± SEM (*n* = 5). * *p* < 0.05, compared with the DNCB/DFE-induced group. Dexa: dexamethasone.

**Figure 3 molecules-26-04409-f003:**
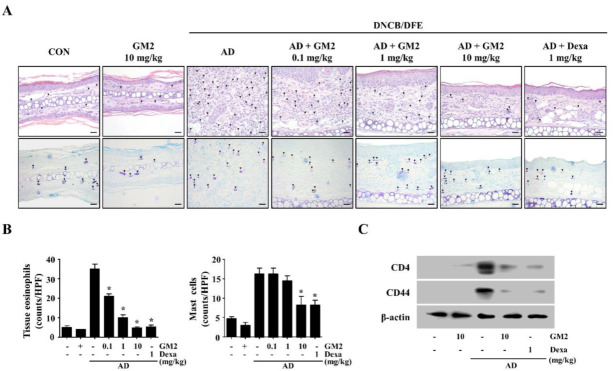
Effect of GM2 on the infiltration of the inflammatory cells in AD-like skin lesions. Representative ear tissue samples from the DNCB/DFE-induced mouse model. (**A**) The infiltration of tissue eosinophils is detected by H&E stain (upper panel, black triangle) and infiltration of the mast cells is detected by TB stain (down panel, black triangle). All scale bar = 20 μm. magnification 400×. (**B**) The number of tissue eosinophils and mast cells is expressed as the mean number of cells at five random sites on each slide. Data are presented in the graph are the means ± SEM (*n* = 5). * *p* < 0.05, compared with the DNCB/DFE-induced group. (**C**) Inhibition of CD4 and CD44 proteins in DNCB/DFE-induced mice ear tissue was analyzed by Western blot analysis. The loading control was confirmed using β-actin in independent blots from three randomly selected ear tissue per each group. Dexa: dexamethasone.

**Figure 4 molecules-26-04409-f004:**
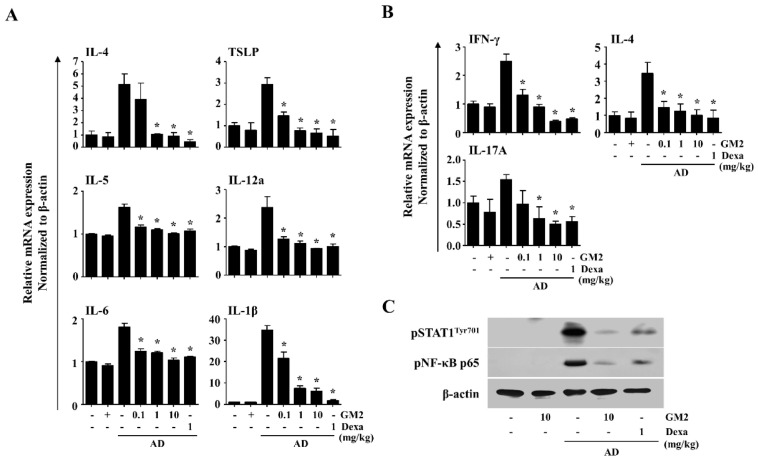
Effects of GM2 on the inflammatory mediator response in AD mice. (**A**) qPCR was used for the measurement of AD-related cytokines in AD-like skin lesions. (**B**) AD-related cytokines were measured by qPCR in the auricular lymph node of AD mice. Data presented in the graph represents the means ± SEM (*n* = 5). * *p* < 0.05, compared with the DNCB/DFE-induced group. (**C**) The activation of STAT1 and NF-κB phosphorylation were analyzed by Western blot in AD-like skin lesions. β-actin was used as a loading control and blots used independent blots from randomly selected three of ear tissues per each group. Dexa: dexamethasone.

**Figure 5 molecules-26-04409-f005:**
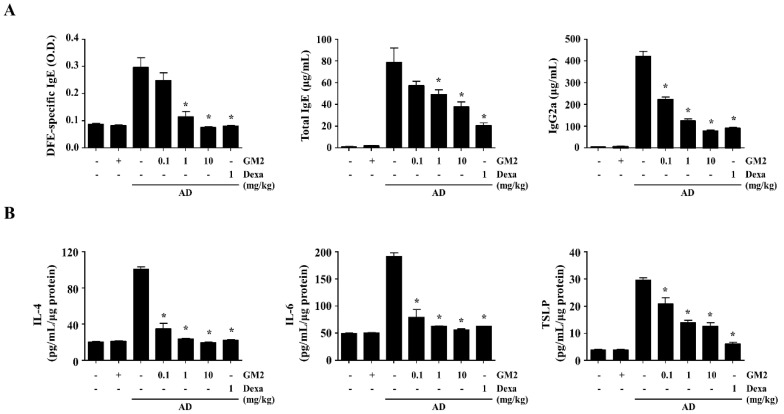
Effect of GM2 on the levels of sera immunoglobulin and tissue extracts of AD mice. Blood samples were collected from the abdominal vena cava and ear tissue samples were removed per mouse from each group on day 28. Each graph demonstrates the levels of DFE-specific IgE, total IgE, and IgG2a in serum (**A**) and the levels of IL-4, IL-6, and TSLP proteins in ear tissue extracts (**B**) were determined using ELISA. Data presented in the graph represents the means ± SEM (*n* = 5). * *p* < 0.05, compared with the DNCB/DFE-induced group. Dexa: dexamethasone.

**Figure 6 molecules-26-04409-f006:**
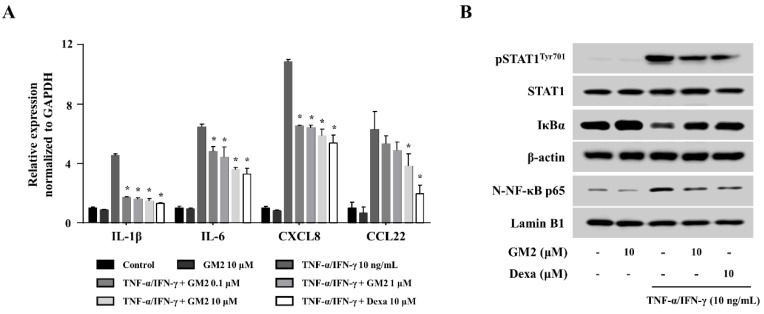
Effects of GM2 on keratinocyte activation. (**A**) Inflammatory cytokine gene expression was measured by qPCR. HaCaT cells (2 × 10^5^ cells/24-well plates) were pretreated with or without GM2 or Dexa for 1 h and subsequently treated with TNF-α/IFN-γ for 6 h. (**B**) Western blot analysis was utilized to evaluate the activation of signaling proteins (p: phosphorylated; N: nuclear). For protein analysis, HaCaT cells (1 × 10^6^ cells/6-well) were pretreated with or without GM2 or Dexa for 1 h and then treated with TNF-α/IFN-γ for 15 min. The STAT1, β-actin and lamin B1 band was used as a loading control. Data presented in the graph represents the means ± SEM (*n* = 3). * *p* < 0.05, compared with the TNF-α/IFN-γ-stimulated group. Dexa: dexamethasone.

## Data Availability

Not applicable.

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
