# Peer review of "Gomisin M2 Ameliorates Atopic Dermatitis-like Skin Lesions via Inhibition of STAT1 and NF-κB Activation in 2,4-Dinitrochlorobenzene/Dermatophagoides farinae Extract-Induced BALB/c Mice"

_molecules, 2021, doi:10.3390/molecules26154409_

Round 1

Reviewer 1 Report

The article titled “Gomisin M2 Ameliorates Atopic Dermatitis-likes Skin Lesions via Inhibition of STAT1 and NF-κB Activation in 2, 4-dinitro-chlorobenzene/ Dermatophagoides farinae Extract-Induced BALB/c mice” is of interest to the scientific community and in general well designed. Since the editor asked me to review the western blot original images, I have developed some concerns specific to the western blot analysis.

  • To look at original data, there should be three separate experiments to confirm reproducibility. Not one representative image for each protein.
  • From what I can tell.... There are bubbles in the CD44 and pNFKB images very close or on the expected protein.
  • When considering that these images with bubbles are the ones authors chose as representative images makes me wonder if three separate experiments were conducted and if the results are truly reproducible.
  • Because of the air bubbles in the region where the proteins of interest should show up, the authors should use ponceau S staining in order to show that the air bubbles do not interfere with their analysis and the area containing their protein of interest did indeed transfer correctly.
  • I may be mistaken, but it seems that Actin controls are done on different membranes than NFKB and CD44 as well.
  • When showing nuclear localization. The cytoplasmic fraction should also be shown along with Tubulin control.

Significant English editing is required. A sentence such as: “GM2 also inhibited STAT1 and NF-κB phosphorylation in DNCB/DFE-induced AD-like lesions. The IgE (DFE-specific and total) and IgG2a in the mice sera reduced as was IL-4, IL-6, and TSLP in ear tissue by oral administrating GM2.” Is hard to read and understand. The manuscript has grammatical errors throughout and a thorough editing would help rewrite the sentences to be more clear and concise.

The protocol represented and written as figure S1 should be in the main manuscript. This represents some of the most vital information in this article. The reader would want to easily access this protocol. How was GM2 administered? By gavage? What was used as a vehicle for the mouse study? Why was 1mg/kg dexamethasone chosen? That seems like a very high dose.

For analysis purposes, it states that n=5. Is that tissue from 5 separate mice or was it replicates of another form? I would like to see the data for all five replicates for both figure 3 and 4.

In Figure 4, beta actin is saturated and there are still clear differences in the amount loaded. I would like to see the western blots where the loading control signals are not saturated.

I do not see methods describing how the mRNA and protein samples were processed from the animal experiment.

Is Figure 6C mislabeled? The quantification is labeled with control PSO GM2 and Dexa, but the pictures are labeled with live cells, apoptotic cells, gm2 and dexa. Does this mean that the images are not representative of the quantification? Why was PSO chosen? What does apoptosis have to do with the authors hypothesis and animal study? Apoptosis was not measured in the animal samples and seems irrelevant by itself in the cell study.

While I do have a lot of concerns about this manuscript, I do think the overall paper is well designed. It does require intense revisions, in both writing and data representation.

Reviewer 2 Report

In their study, the authors present evidence for the anti-inflammatory effects of GM2, a lignan extracted from Schisandra chinensis, in a  mouse model for atopic dermatitis. They show GM2 can suppress inflammatory cytokines in a very similar manner to dexamethasone, acting on STAT1 and NF-kB. Similar effects  had been shown previously for an entire Schisandra extract.

While the manuscript is generally straightforward and the results are  mostly presented  quite clearly, I have a few remaining questions and suggestions.

1) The authors should take care to check their manuscript for missing words and grammatical issues that alter the meaning of the respective sentences. For instance, in the very first sentence of the abstract, the authors probably want to state that Schisandra extracts have been shown to have beneficial effects on allergy and inflammation and not that their effects include allergy and inflammation.

2) I am not sure that thee assumption that plant compounds are generally safe and don't have side effects. As the effects of GM2 are similar to those of corticosteroids, in my opinion similar side effects cannot be excluded - unless the authors can cite long-term studies that really prove this.

3) The experiment with the HaCaT cell line and  the apoptosis assay is not quite clear to me. Did the authors treat the cells with the apoptosis inducer with or without GM2 or Dexamethasone, and GM2 lowered the apoptosis induction by the psoralidine, as suggested by the text? Or did they treat the cells with one of these substances only and both GM2 and Dexamethasone also induced apoptosis to some extent, but to a lesser degree than psoralidine? What is the significant change indicated by the asterisk?

4) The discussion is a bit redundant, because it basically recapitulates eveery single result. This section could be shortened significantly.
